# Japanese quail (*Coturnix japonica*) selected for high social motivation rely on conspecifics for buffering but not learning in sociocognitive tasks

Jeanne Seressia[1], Alexia Morel[1], Fabien Cornilleau[1], Julie Lemarchand[1], Léa Lansade[1], Christian Nawroth[2], Ludovic Calandreau[1‡]*, Vitor Hugo Bessa Ferreira[1‡]*

**1** INRAE, CNRS, IFCE, Université de Tours, Centre Val de Loire UMR Physiologie de la Reproduction et des Comportements, Nouzilly, France, **2** Research Institute for Farm Animal Biology (FBN), Institute of Behavioural Physiology, Dummerstorf, Germany

‡ These author co-last authorship on this work.
* jeanne.seressia@inrae.fr (JS); vitor.ferreira@inrae.fr (VHBF)

## Abstract

Social living is widely considered a key driver of cognitive evolution, yet individuals within a species and even within the same group can differ substantially in their sociability (i.e., an individual's propensity to form and maintain social bonds), which can ultimately shape the social environment they experience by influencing how they interact with, respond to, and engage in it. How such individual variation in this personality trait affects social cognition, however, remains poorly understood. To address this question, we used two selectively bred lines of Japanese quail (*Coturnix japonica*) that consistently differ in social motivation, a key component of sociability, which we used as a proxy for this broader trait. In these lines, S+ individuals show high social motivation, whereas S- individuals show low social motivation. We compared their sociocognitive performance across three complementary tasks: a gaze following task, a social buffering task, and a social discrimination learning task. Our findings revealed that Japanese quail reliably followed the gaze of conspecifics, providing the first evidence of this ability in this species. However, there was no difference between lines, suggesting that basic gaze following into the distance is independent of social motivation. In contrast, line differences emerged in the other tasks. S+ quail were more sensitive and less adaptable in response to environmental changes, particularly under social isolation, although the presence of a conspecific strongly buffered these effects. S- quail, on the other hand, outperformed S+ individuals in the social discrimination learning task, rapidly exploiting available social cues to guide foraging decisions. Overall, this study demonstrates that social motivation modulates sociocognitive performance in a context-dependent manner. Rather than conferring a general cognitive advantage, divergent selection on social motivation leads to different strategies of social information use across tasks and contexts, highlighting the complex interplay between personality, social environment, and cognition.

**Data availability statement:** All relevant data are within the paper and its Supporting Information files.

**Funding:** This work was supported by the PHASE Research Department (INRAE) and a doctoral fellowship granted to Jeanne Seressia by the University of Tours, France.

**Competing interests:** The authors have declared that no competing interests exist.

## Introduction

Group living confers major adaptive benefits, including enhanced protection against predators, improved access to resources, and increased reproductive opportunities [1,2]. However, it also entails substantial cognitive demands, as individuals must navigate complex and dynamic social environments, manage repeated interactions with conspecifics, and maintain cohesive, stable relationships over time, requiring a broad array of abilities such as social recognition, communication, cooperation, and social learning [3,4]. According to the Social Intelligence Hypotheses, such social challenges have driven the evolution of advanced cognition, a view primarily supported by interspecific comparative studies linking social complexity to brain size and cognitive performance [4–7].

While sociality is well established as a key factor in the evolution of cognitive abilities across animal species, far less attention has been devoted to how social environments are experienced at the individual level [6,8,9]. Importantly, individuals within the same species, and even within the same group, may be exposed to markedly different social conditions. This growing recognition of intraspecific variation in social environments has fuelled a surge of interest in how factors such as group size, group stability, social network position and social rank relate to individual cognitive performance [10–16]. For instance, individuals living in larger, dynamic groups, or occupying more central or low-ranking social positions may experience greater social complexity and, consequently, higher cognitive demands [4,6,11–14,17]. However, evidence for these relationships remains scarce and equivocal, suggesting that additional social drivers may contribute to intraspecific variation in cognitive performance.

Importantly, variation in social experience may extend beyond externally defined social structures. Individuals may also differ intrinsically in their motivation to interact with conspecifics and to establish social relationships, thereby actively shaping their effective social environment [1,3,8,18,19]. The above consideration point to a critical source of phenotypic variation: individual personality. Personality refers to consistent behavioural differences among individuals across time and contexts, with sociability representing one of its principal axes [20–23]. Sociability can be defined as an individual's propensity to form and maintain social bonds [11,20,24]. Within a given group, individuals can exhibit important differences in their level of sociability, with some actively seeking social interactions, and others adopting a more independent or avoidant disposition [1,4,19,24]. Such variation is likely to translate into differences in the cognitive demands individuals face, for example by influencing the number of interactions or differentiated relationships they experience and the ways in which they engage in them, as demonstrated for other personality traits, notably exploration and boldness, whose links with cognition have been extensively documented [25–28]. Yet, despite sociability being a central axis of animal personality with potentially wide-ranging socioecological repercussions, its connections to cognitive processes have received very little empirical attention to date [1,8,18,19].

To narrow this knowledge gap, the present study relied on two unique selectively bred lines of Japanese quail that differ markedly in social motivation: the S+ line, characterised by high social motivation, and the S- line, characterised by low social

motivation [29]. Social motivation refers to an individual's internal drive to seek contact with, move towards, and remain in proximity to conspecifics [30]. As an important component of sociability underlying most social interactions, it provides an interesting proxy for this broader trait, and we therefore used it as such in this study. Moreover, divergence in social motivation between S+ and S- quail has been shown to be robust, persisting into adulthood and across a wide range of experimental contexts [29–33]. Selection for social motivation has also produced broader differences in sociability-related behaviours. For example, S+ birds maintain shorter inter-individual distances at an early developmental stage [34], and display non-specific social attraction to peers whereas S- quail preferentially orient towards familiar conspecifics [35–37]. S+ individuals also exhibit higher behavioural synchronisation, particularly during resting behaviours [38]. These behavioural differences are mirrored at the neurobiological level, through differential expression of key neuropeptides involved in social behaviours, such as mesotocin and vasotocin [39]. Together, these characteristics make these quail lines a valuable model to investigate how variation in social motivation, and sociability more broadly, influence cognitive abilities. In addition, Japanese quail themselves represent a particularly relevant model for investigating this question due to their rich social behavioural repertoire and social organisation, the high level of experimental control achievable both before and after hatching, and the fact that their sociocognitive abilities remain comparatively underexplored relative to those of more extensively studied avian models, such as corvids [40–42].

To this end, we employed in this study three complementary sociocognitive tasks, each targeting distinct aspects of social cognition. First, a gaze following task assessed the individual's propensity to look in the same direction others are looking and use the attentional cues they provide. Second, a social buffering task evaluated the individual's propensity to rely on the presence of conspecifics to attenuate stress responses. Third, a social discrimination learning task tested the individual's ability to associate and discriminate between a social stimulus and a physical stimulus. Social cognition is not considered a unitary ability but rather a suite of skills that can be organised according to the cognitive demands they place on individuals [3]. Within this framework, the three tasks used here collectively probe social cognition from multiple perspectives, that vary in both the type of social information processed and the context in which it is used. Gaze following represents one of the most fundamental sociocognitive abilities, enabling individuals to rapidly respond to socially relevant information from others' attentional cues. It reflects a largely reflexive sensitivity to social signals and was assessed here in a neutral context [43]. Social buffering captures the integration of social information into emotional regulation processes. By measuring the modulation of stress responses depending on the social environment, this task targets a distinct dimension of social cognition linking the perception of conspecifics to physiological regulation [44]. Finally, social discrimination learning engages higher-level integrative cognitive processes, requiring individuals to form and use learned associations involving social cues in a goal-directed, feed-motivated context. Together, these paradigms allow us to test whether individual variation in sociability, assessed using our divergent quail lines, is linked to changes in social cognition as a whole or only in specific, partially independent components. We predicted that S+ and S- quail would differ in their performance across these tasks. Specifically, we expected S+ individuals to attend more closely to social information and social context, given their higher social motivation and, consequently, greater engagement with conspecifics. We hypothesized that S+ individuals would be more likely to follow the gaze of conspecifics, show greater modulation of stress responses by social context, and rely more readily on social information, resulting in faster association in the social discrimination learning task.

## Materials and methods

### Ethics statement

This study was conducted at the Pôle d'Expérimentation Avicole de Tours (UE PEAT, INRAE, Experimental Poultry Facility) of the INRAE Centre Val de Loire, France, from March to July 2025. The protocol was approved by the INRAE Animal Experimentation Ethics Committee (project number #CE19-2024-2511-1), in compliance with the current French legislation and the European Directive 2010/63/EU on the protection of animals used for scientific purposes. All procedures

were designed to minimise potential adverse effects on animal welfare. Birds were housed in wood-shaving-bedded pens enriched with mesh tubes, and maintained in stable social groups throughout the study. Capture and handling were carried out gently by trained personnel. Birds were removed from their home pens only for routine husbandry procedures and brief testing sessions, after which they were promptly returned to their familiar environment. Animal welfare was monitored daily for signs of discomfort, injury, or illness. Adult quail body weight was recorded every two weeks and remained stable throughout the study, confirming that the feed-restriction schedule did not induce weight loss or compromise welfare. All females involved in the experiments were responsibly rehomed to small farms and private adopters. After sexing, a substantial proportion of males were also successfully rehomed despite the known challenges associated with placing male Japanese quail. The remaining males were euthanised by cervical dislocation, a standard method that induces instantaneous loss of consciousness and death, thereby minimising suffering. All euthanised animals were subsequently repurposed as feed for wildlife breeders or organisations keeping falcons or reptiles.

## History of lines

The experiments were conducted on the 78th generation of a model of two lines of Japanese quail, selected for divergent levels of social motivation since 1984 at the Pôle d'Expérimentation Avicole de Tours. Selection was originally performed using a treadmill test until the 45th generation. This test was conducted to identify individuals at the extremes of social motivation: those with the highest social motivation were used as breeders for the S+ line in the following generation, while those with the lowest social motivation were used as breeders for the S- line. Then, from the 46th generation onwards, the selection test was no longer performed and breeders were chosen randomly within each line to produce the next generation. However, the treadmill test has been carried out approximately every three years on a random sample of birds from both lines to ensure that the divergence was maintained. The quail used in the present study did not participate in the treadmill test.

Details on the selection procedure implemented during the initial generations are described in Mills and Faure [29]. Briefly, during the treadmill test, individual chicks aged 6–8 days are placed in the centre of a corridor, with five conspecifics from a control line (ST) positioned at one end inside a box, as social stimuli. When a chick moves towards the conspecifics, the treadmill is activated, gently pulling the chick away. Over a 5-minute period, the distance travelled on the treadmill by the chick attempting to reach its conspecifics and the time spent at the opposite end are measured. Chicks that travel the greatest distances and spend the least time away from conspecifics are classified as having high social motivation (S+), whereas those that travel the shortest distances and spend the most time away are classified as having low social motivation (S-). The selection procedure ensured that social motivation was independent of tonic immobility response, a classic behavioural indicator of fear in birds, indicating that the high S+ and low S- lines should not differ in baseline fear levels [29].

## Subjects and housing

All birds used in this study were incubated and hatched in our experimental facilities, at the Pôle d'Expérimentation Avicole de Tours. Immediately after hatching, chicks retained for the study were fitted with a unique numbered wing tag for individual identification, and then transferred to a dedicated rearing facility.

From 1 to 21 days of age, 308 chicks, including 120 S+, 120 S-, and 68 ST, were housed in a single room divided into two pens (each pen measuring 500 × 180 × 250 cm). The S+ birds were placed in one pen and the S- birds in the other, to avoid cross-line social influence [39]. The pens were separated by opaque plastic panels (150 cm high) to prevent visual and physical interactions. ST birds were randomly and evenly distributed between the two pens. In each pen, the floor was covered with wood shavings, and the birds had *ad libitum* access to feed (one large feed hopper, 30 cm in diameter and 30 cm in height) and water (10 automatic drinking pipettes, adjustable according to the birds' size). Several plastic mesh tubes were distributed on the ground to provide environmental enrichment. The ambient temperature was initially

set at 40°C on day 1, gradually lowered to 24 °C by day 21, and maintained at this average temperature thereafter. Lighting began with continuous artificial light (24h) on day 1 and progressively adjusted to a 16h light (8:00 am to midnight)/8h dark cycle.

At 22 days of age, once males and females could be distinguished based on feather dimorphism, the birds were sexed. Only females were retained for this study, as males become aggressive towards each other and frequently display sexual behaviour towards females once they reach sexual maturity at around six weeks of age [34].

After sexing, a total of 118 females were randomly chosen: 46 S+, 44 S-, and 28 ST (13 in the S+ pen and 15 in the S- pen). This resulted in 59 quail per pen, corresponding to a density of approximately 6.5 birds/m². In addition to their wing tags, these individuals were marked with coloured metal leg rings (6 mm diameter) to allow quick and easy identification. To habituate the animals to human presence, experimenters spent 15–30 minutes daily in each pen prior to the start of testing.

## Cognitive tasks

Three sociocognitive tasks were conducted: a gaze following task, a social buffering task, and a social discrimination learning task. Not all birds participated in all tasks; one group of S+ and S- individuals was assigned to the gaze following task, while a different set of individuals was assigned to the social buffering task followed by the social discrimination learning task. ST individuals were additionally used across tasks as demonstrators or social cues.

The social buffering task and the social learning task relied on feed rewards as motivation. Mealworms were used because they are highly appetizing to quail, and they were provided dead for practical reasons and to prevent birds from relying on auditory cues. To ensure that all individuals remained consistently motivated during the experiments and to standardise motivation across them, we implemented a controlled and predictable feed-restriction schedule prior to testing: birds tested in the morning were fasted during the dark phase, when quail do not feed naturally, whereas birds tested in the afternoon received a 30-min feeding period in the morning before being fasted until testing. Water was available *ad libitum*, and no feed restriction was applied for the gaze following task. Half of the S+ and S- individuals were tested in the morning and the other half in the afternoon, with testing alternating between S+ and S- individuals in a randomised order.

Data were collected by two researchers (JS and AM). For each test, behaviours were scored by a single observer to ensure within-test consistency. Prior to data collection, both observers underwent a training period, during which behaviours were discussed and operational definitions were refined until agreement was reached. Data collection was then conducted independently.

**EXPERIMENT 1: Gaze following task.** Inspired by Loretto et al., Nawroth et al., and Zeiträg et al. [45–47], we performed a gaze following task to assess individuals' propensity to use a conspecific's attentional cues by following its gaze into the distance (Fig 1). The task was conducted on 40 focal individuals (20 S+ and 20 S-). In addition, 10 ST individuals served as demonstrators, with 5 housed in the S- pen and 5 in the S+ pen. Focal birds and ST demonstrators were paired randomly, but ensuring that they were not housed in the same pen to prevent any influence of pre-existing social bonds. The pairs remained unchanged throughout the experiment. The task began when the birds were 31 days old and ended when they were 71 days old.

**Experimental setup.** The task was conducted in a square arena (80×80×40 cm) located in a testing room near the rearing room. The walls of the arena were made of white painted wood, the floor was laid with dark green linoleum lined with wood shavings and the arena was covered with a net to prevent the birds from escaping. The arena was divided into two equal compartments (80×40×40 cm) by a wire mesh, with one compartment for the focal individuals and the other for the ST demonstrators. A small panel (20×10 cm) made of white expanded PVC was attached to the centre top of the mesh on the ST demonstrators' side to serve as a surface for projecting a visual stimulus (red laser). Two cameras (Panasonic HC-VX980 and HC-VX870) were positioned on either side of the arena, to record trials from multiple angles. The experimenter was also present near the arena during all trials, standing on the ST demonstrators' side.

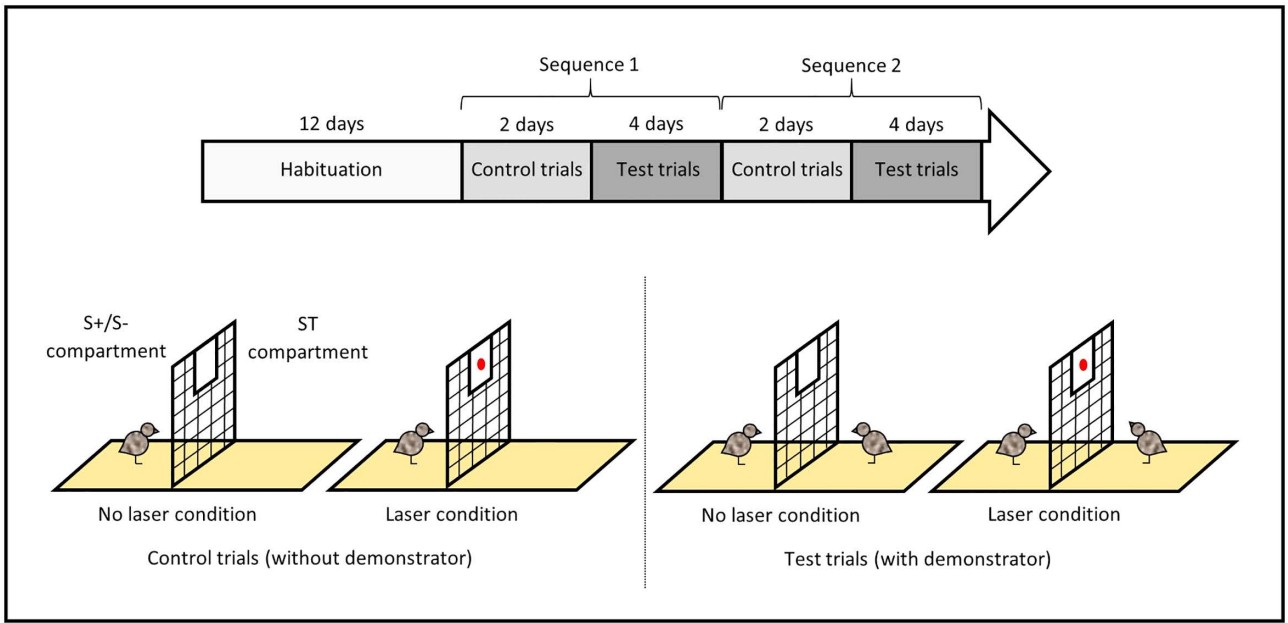

**Fig 1. Timeline of the experimental procedure and schematic representation of the gaze following task.**

**Procedure.** First, the focal S+ and S- birds and the ST demonstrators were habituated to the arena and its components over 12 days (Fig 1). On the first day, birds were habituated in groups of 10, consisting of five focal individuals from the same line placed in one compartment of the arena and their five corresponding ST demonstrators placed on the opposite side. During the following five days, birds were habituated in pairs, with each focal bird tested together with its assigned demonstrator, each positioned on their respective side of the arena. Over the next four days, only focal birds were tested individually, alone on their side of the arena. Finally, during the last two days, birds were tested again in pairs to re-engage the ST demonstrators with the task. On the first day each habituation trial lasted 30 minutes, and 5 minutes on the following days. This procedure ensured a gradual transition to social isolation. Habituation was considered successful when all individuals were able to remain calm for most of the trial, showing non-vigilant, foraging behaviour.

Once habituation was achieved, control trials took place, during which focal birds were tested alone without demonstrator (Fig 1). Control trials were conducted to ensure, prior to the test trials (see paragraph below), that focal birds were not able to detect the presentation of a red laser projected onto the panel on the demonstrators' side. All individuals completed four consecutive trials per day over two days, alternating between two conditions: with laser presentation (laser condition) and without laser presentation (no laser condition). Each bird therefore completed eight control trials in total, four per condition. A trial began once the bird was oriented towards the mesh. In the laser condition, the laser was projected onto the panel on the ST demonstrator's side for 5 seconds, and the trial ended after an additional 10 seconds. In the no laser condition, the trial began once the bird was correctly positioned and lasted 10 seconds.

Then, test trials for gaze following assessment occurred, during which focal birds were tested with their assigned ST demonstrator (Fig 1). Each day during four days, half of the individuals completed four consecutive trials, alternating between two conditions: with laser presentation (laser condition) and without laser presentation (no laser condition). Each bird therefore completed eight test trials in total, four per condition. A trial began once both the focal bird and the ST demonstrator were oriented towards the mesh and able to see each other. In the laser condition, the laser was projected onto the panel on the ST demonstrator's side until the demonstrator raised its head towards the panel, and the trial ended

after an additional 10 seconds. It should be noted that the height of the panel was deliberately chosen so that the ST demonstrator had to lift its head to inspect more closely the laser dot. In the no laser condition, the trial began once the birds were correctly positioned and lasted 10 seconds.

The sequence of two days of control trials followed by four days of test trials was repeated a second time in the exact same way to increase trial numbers and ensure consistency (Fig 1).

**Recorded behavioural variables.** For each control trial without demonstrator, it was recorded whether the focal bird raised its head at least once towards the panel (positive response), or did not (negative response), during the two conditions without (10 seconds) or with laser presentation (5 seconds of laser presentation followed by 10 seconds).

For each test trial with demonstrator, it was recorded whether the focal bird raised its head at least once towards the panel or towards the wall behind—since it could interpret the demonstrator's gaze as being directed to a location behind rather than above itself [47]—(positive response), or did not (negative response), during the two conditions without (10 seconds) or with laser presentation (10 seconds following the demonstrator's gaze).

More specifically, a head raise in a positive response was defined as any instance within the trial's observation window in which the focal bird tilted its head towards the panel, or towards the wall behind in test trials with a demonstrator, and/or markedly elevated its head (Fig 2). Only clear head movements were scored as positive responses.

**EXPERIMENT 2: Social buffering task.** Inspired by Stenfelt et al., and Gjøen et al. [48,49], we performed a social buffering task to assess individuals' response to an environmental disturbance (here, a change in the colour of a feeder and its surroundings after birds had been habituated to it) depending on whether they were alone or accompanied by a conspecific (Fig 3). The task was conducted on 40 focal individuals (20 S+ and 20 S-). In addition, 10 ST individuals served as demonstrators, with 5 housed in the S- pen and 5 in the S+ pen. Half of the focal birds was tested in pairs with a ST demonstrator (paired condition) from the opposite pen, and the other half was tested alone (alone condition). The condition was randomly and evenly assigned to S+ and S- birds, and for the paired condition, focal-demonstrator pairings were also randomised and remained unchanged throughout the experiment. The task began when the birds were 31 days old and ended when they were 92 days old.

**Experimental setup.** The task was conducted in an arena (100 × 100 × 50 cm) located in a testing room near the rearing room. The walls of the arena were made of white expanded PVC, the floor was laid with dark green linoleum and the arena was covered with a net to prevent the birds from escaping. A feeder (30 × 5 × 5 cm) containing the birds' daily pellets and mealworms was positioned in the arena. A camera (Sony HDR-CX405), mounted above the arena and connected to a screen, recorded the trials and allowed the experimenter to follow the animals from a distance.

**Procedure.** ST demonstrators were first habituated to the arena and to three feeder configurations differing in colour (orange, black, and white) over six days (Fig 3). Each individual completed three consecutive trials per day (one per feeder), for a total of 18 trials. Habituation proceeded from social to individual testing: birds were tested in groups of five from the same pen on day one, in groups of two or three on day two, and individually thereafter. Trial duration was progressively reduced, lasting 10 minutes on day one, 5 minutes on day two, and ending thereafter once the bird began eating from the feeder. By the end of habituation, all ST demonstrators reliably fed from each feeder within one minute, ensuring familiarity with all experimental configurations.

Focal S+ and S- birds were then habituated to the arena and to a single feeder configuration (orange) over seven days (Fig 3). Each individual completed two consecutive trials per day, for a total of 14 trials. On day one, birds were tested in groups of five from the same line, in pairs on day two (either with their assigned ST demonstrator in the paired condition or with a conspecific from the same line in the alone condition), and thereafter either in pairs (paired condition with their ST demonstrator) or individually (alone condition). Trial duration followed the same reduction as for ST demonstrators. In the paired condition, the ST demonstrator was always placed in the arena first and allowed to begin feeding before the focal bird was introduced. It should be noted that the feeder was sufficiently long to ensure that the demonstrator could not monopolise access to it or obstruct the focal bird's view of its colour. By the end of habituation, focal birds fed from the

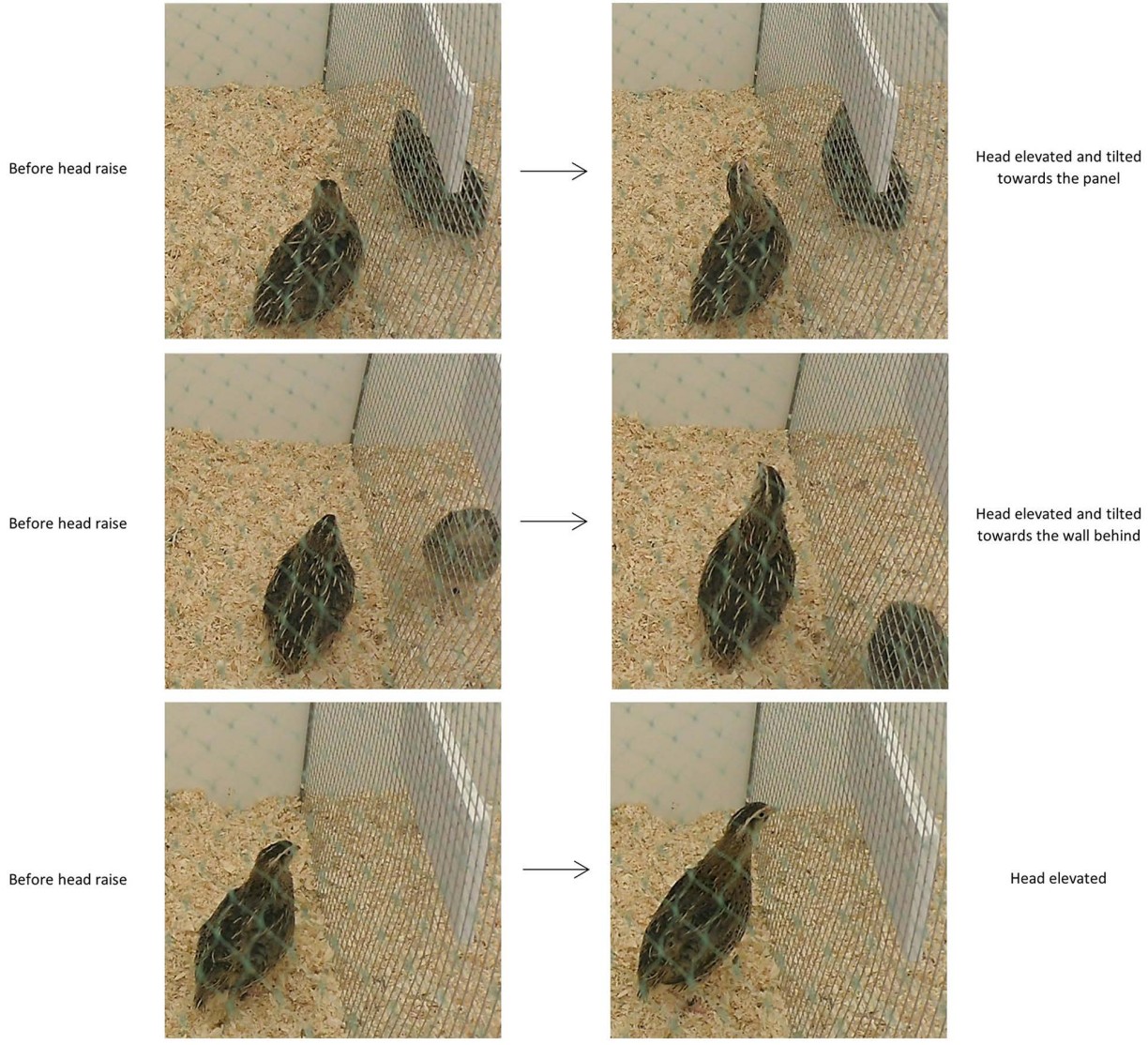

Before head raise → Head elevated and tilted towards the panel

Before head raise → Head elevated and tilted towards the wall behind

Before head raise → Head elevated

**Fig 2. Representative images illustrating examples of head raises scored as positive responses according to the behavioural coding protocol.**

orange feeder in less than one minute. During this phase, ST demonstrators also completed two daily trials alone with the black feeder to maintain familiarity with it.

Testing took place the day after the final habituation session (Fig 3). Each focal bird completed two consecutive trials in its assigned condition (paired or alone). The first trial used the familiar feeder (orange) to establish a baseline, followed by a second trial with a novel, unfamiliar feeder (black) to assess social buffering. Trials ended when the focal bird reached the feeder and placed its head above it, or after a maximum of two minutes if it failed to do so.

Following the first test session, focal birds were habituated to the black feeder using the same procedure as above, except that initial group habituation trials were omitted as birds were already accustomed to social isolation (Fig 3). ST demonstrators completed two supplementary daily trials alone with the white feeder to maintain familiarity with this

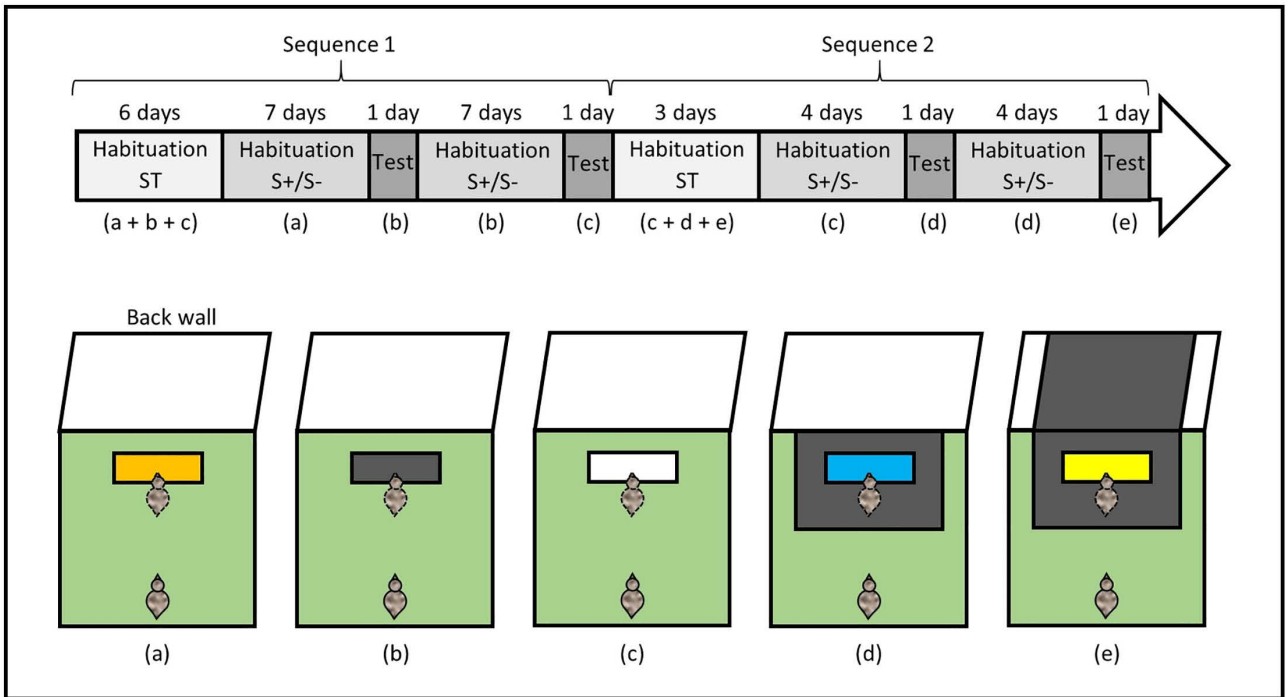

**Fig 3. Timeline of the experimental procedure and schematic representation of the social buffering task.** Letters (a–e) correspond to the different experimental setups illustrated below the timeline. The bird outlined with a dashed line in the feeder zone represents the ST demonstrator and was present only in the paired condition. In experimental setups (c) and **(d)**, the black rectangles represent a sheet of thick paper (75 × 50 cm) added to modify the visual surroundings of the feeder.

configuration. A second test session then followed, identical in structure to the first, with the black feeder serving as the familiar feeder and the white feeder as the unfamiliar feeder.

The entire experimental sequence (ST demonstrator habituation; focal S + /S- habituation; test; focal S + /S- habituation; test) was repeated a second time to increase reliability across line and condition subgroups (Fig 3). In this replication, both feeder colour and surrounding context (floor beneath and wall behind the feeder) were modified to prevent habituation to feeder colour changes alone. After habituation to the familiar white feeder, birds were thus exposed to an unfamiliar blue feeder presented on a black floor, then habituated to this configuration and tested with an unfamiliar yellow feeder presented on a black floor in front of a black wall. ST demonstrators were habituated in advance to all configurations until they reach the feeder in less than one minute. Habituation phases were shortened (three days for ST demonstrators and four days for focal birds), as all birds were already familiar with social isolation and configuration changes were more salient.

**Recorded behavioural variables.** On each of the four testing days, during the first trial with the familiar feeder or configuration, the latency (in seconds) for focal birds to reach the feeder and place the head above it was recorded. During the second trial with the unfamiliar feeder or configuration, the same latency was recorded and set to 120 seconds if the subject failed to reach the feeder within two minutes.

**EXPERIMENT 3: Social discrimination learning task.** Inspired by Zidar et al. [28], and Seressia et al. (submitted), we performed a social discrimination learning task to assess individuals' ability to associate and discriminate between two stimuli (here, the location of a feed reward and the presence of a nearby conspecific, Fig 4). The task was conducted on 37 focal individuals (19 S+ and 18 S-). In addition, 9 ST individuals served as social cues, with 5 housed in the S- pen and

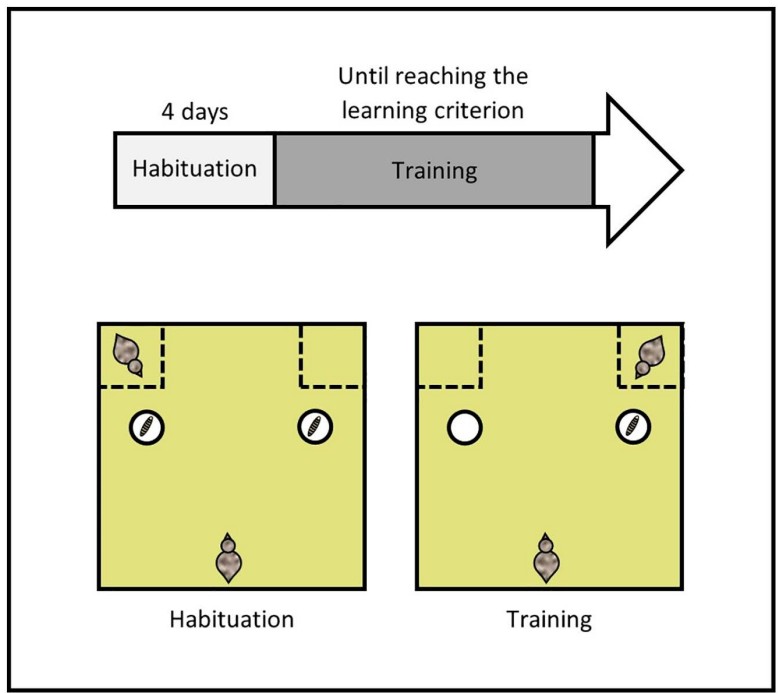

**Fig 4. Timeline of the experimental procedure and schematic representation of the social discrimination learning task.**

4 in the S+ pen. These individuals were the same ones that had previously participated in the social buffering task. Focal and ST birds were paired randomly, while still not being housed in the same pen and with the constraint that the pairs differed from those constituted in the social buffering task. The pairs remained unchanged through the experiment. The task began when the birds were 100 days old and ended when they were 128 days old.

**Experimental setup.** The task was conducted in the same arena as the social buffering task, but the floor was changed to a light green/yellow linoleum, in order to vary the experimental conditions and limit the potential influences between tasks. Two small wire-mesh compartments (25×20×50 cm) were positioned in the corners of one side of the arena, with two white ceramic cups spaced 40 cm apart placed in front of them. A camera (Sony HDR-CX405), mounted above the arena and connected to a screen, recorded the trials and allowed the experimenter to follow the animals from a distance.

**Procedure.** First, the focal S+ and S- birds were individually habituated to the arena and its components over four days (Fig 4). Each individual completed four consecutive trials per day, totalling 16 trials. Additionally, for the animals to learn they could eat the mealworms from both cups, two mealworms were placed on the floor in front of each cup and two mealworms were placed inside each cup during the initial habituation trials. By the end of the habituation, this was progressively reduced so that only one mealworm remained inside each cup. One ST bird was consistently present in one of the two wire-mesh compartments, with its position alternated between each trial. A trial ended once the individuals consumed all available mealworms. Habituation was considered successful when the majority of the individuals were able to eat the mealworm from both cups within one minute.

Discrimination training started after the focal birds finished their habituation (Fig 4). Each individual completed eight consecutive training trials per day. During these trials, one mealworm was placed in only one of the two cups, always on the side as the compartment containing the ST bird. To ensure that the animals learned to locate the feed in the cup indicated by the ST bird positioned behind it, the position of the rewarded cup and the ST bird were randomized, with no

more than two consecutive trials on the same side. A trial ended once the focal individual consumed the mealworm, even if it visited the wrong, empty cup first, in order to keep it motivated throughout training. If the bird failed to eat the mealworm within one minute, the trial was stopped and the bird was gently guided towards the correct cup. An individual was considered to have learned the task and met the learning criterion when it was able to immediately locate the mealworm in the correct cup, without first visiting the incorrect cup, across five consecutive trials over two consecutive days. It should be noted that the cups were opaque and designed so that the quail could not see the mealworm until they had their head directly over the cup. An individual was thus considered to have visited a cup when it moved its head directly above it. Additionally, to decrease the likelihood that smell guided the choice of the quail, one mealworm was also placed in the second cup but hidden beneath a false bottom, therefore invisible to the bird.

**Recorded behavioural variables.** For each training trial, it was recorded whether the focal bird visited the correct cup first, or the incorrect one. Additionally, its latency (in seconds) to visit the correct cup and consume the mealworm was recorded.

## Statistical analysis

Statistical analyses were performed using R software version 4.4.2. For all models presented below, non-significant interactions were removed, retaining only main effects. When significant effects were found, post hoc ANOVA comparisons of estimated marginal means were performed as appropriate, using the emmeans package [50] with Tukey adjustment to control for multiple comparisons. We verified model assumptions, including the normality of residuals and homogeneity of residual variance, through graphical evaluations (using the DHARMa package [51]). α level was set at 0.05. Results are reported in the text as mean ± standard deviation.

**Gaze following task.** Of the initial 40 individuals, one S- individual died from unknown causes during the experiment; its data were therefore excluded from the analyses. As a result, statistical analyses were conducted on 20 S+ and 19 S- individuals. For the control trials, the response was analysed (1 = positive response with head raise, 0 = negative response with no head raise), using a Generalized Linear Mixed Model with a binomial distribution (GLMM, using the glmmTMB package [52]). For the test trials, the same model was used. For these two models, the line (S+ and S-) and the condition (laser and no laser), along with their interaction, were initially included as fixed factors. The individual was included as a random factor to account for repeated measures.

**Social buffering task.** Of the initial 40 individuals, one S+ and one S- individuals (both from the alone condition) died from unknown causes during the experiment; their data were therefore excluded from the analyses. As a result, statistical analyses were conducted on 19 S+ and 19 S- individuals. For the four tests, a difference was calculated by subtracting the latency to reach the familiar feeder from the latency to reach the unfamiliar feeder, in order to quantify the disturbance induced by the feeder change independently of individual baseline latencies. This variable was analysed using a Linear Mixed Model (LMM, using the lmerTest package [53]). The line (S+ and S-), the condition (paired condition and alone condition) and the test (1, 2, 3 and 4), along with their interactions, were included as fixed factors. The individual was included as a random factor to account for repeated measures. The difference was transformed using the inverse hyperbolic sine (arcsinh) transformation, which can accommodate zero and negative values and was identified as the most appropriate normalization method using the bestNormalize package [54]. Based on the results of this analysis (see Results section below), the four tests appeared to generate different levels of disturbance, and strong effects of line and condition were also observed. To obtain an overall measure independent of the test, a complementary analysis was conducted using the mean difference across the four tests, averaged across all four trials performed by each individual. Additionally, to prevent strong main effects from masking a potentially subtle line × condition interaction, a group variable was created to represent the four subgroups (1 = S+ alone, 2 = S- alone, 3 = S+ paired, 4 = S- paired). The mean difference was analysed using a Linear Model (LM), with the group included as a fixed factor.

**Social discrimination learning task.** Of the initial 37 individuals, two S+ and two S- individuals did not pass the habituation phase because they were not consistently able to eat the mealworm from both cups within one minute across habituation trials; their data were therefore excluded from the analyses. Additionally, one S+ and four S- individuals did not reach the predefined learning criterion within the maximum training period allowed (15 days). Some individuals showed a decline in motivation during the training phase, as evidenced by repeated trials in which they did not consume the mealworm from either cup, and others displayed intermittent errors embedded within otherwise correct performance sequences. Although the latter showed partial evidence of learning, they did not meet the stringent, a priori learning criterion. The data from all these individuals were therefore also excluded from the analyses. As a result, statistical analyses were conducted on 16 S+ and 12 S- individuals. The total number of trials required to reach the learning criterion and the mean latency to visit the correct cup, averaged across all trials performed by each individual, were analysed using a Linear Model (LM), with the line included as fixed factor. For latency, the individual mean value was used to ensure comparability across individuals despite unequal numbers of trials, and because latency was considered a complementary measure to the primary outcome variable, namely the total number of trials required to reach the learning criterion.

## Results

**Gaze following task.** For the control trials, as expected, the response did not differ between lines (GLMM, $\chi^2_1 = 3.74$; $p = 0.053$), or between conditions (GLMM, $\chi^2_1 = 0.51$; $p = 0.47$, Table 1, Fig 5). Accordingly, regardless of the line and the condition, birds did not raise their heads more frequently towards the panel, notably indicating that they did not detect the laser stimulus (laser: $14.7 \pm 35.5\%$; no laser: $16.8 \pm 37.4\%$). For the test trials, the response did not differ between lines (GLMM, $\chi^2_1 = 0.00$; $p = 1.00$), but a significant condition effect was found (GLMM, $\chi^2_1 = 23.09$; $p < 0.001$, Fig 5). Birds raised their heads towards the panel or towards the wall behind more frequently when the laser was presented than when it was absent (laser: $37.8 \pm 48.6\%$; no laser: $20.3 \pm 40.3\%$).

**Social buffering task.** The difference between the latency to reach the familiar feeder and the latency to reach the unfamiliar feeder significantly differed between lines (LMM, $F_{1,35} = 5.50$; $p = 0.02$), conditions (LMM, $F_{1,35} = 45.26$; $p < 0.001$), and tests (LMM, $F_{3,110} = 20.86$; $p < 0.001$, Table 1, Fig 6). S+ individuals took longer to reach the unfamiliar feeder than S- individuals (S+: $38.9 \pm 47.2$ s; S-: $21.1 \pm 31.5$ s), paired individuals reached the unfamiliar feeder more rapidly than alone individuals (paired: $18.2 \pm 34.7$ s; alone: $42.8 \pm 43.4$ s), and the latency to reach the unfamiliar feeder was shorter during test 2 compared with tests 1, 3 and 4 (test 1: $34.2 \pm 42.6$ s; test 2: $4.8 \pm 8.7$ s; test 3: $54.1 \pm 48.0$ s; test 4: $26.8 \pm 36.8$ s). The mean difference across the four tests significantly differed between groups (LM, $F_{3,34} = 11.64$; $p < 0.001$, Fig 6). Group 1 (S+ alone) took longer to reach the unfamiliar feeder than groups 2, 3 and 4 (group 1/S+ alone: $57.1 \pm 21.5$ s; group 2/S- alone: $28.5 \pm 13.4$ s; group 3/S+ paired: $22.3 \pm 16.5$ s; group 4/S- paired: $14.5 \pm 14.1$ s).

**Social discrimination learning task.** A significant line effect was found both in the total number of trials required to reach the learning criterion (LM, $F_{1,26} = 15.71$; $p < 0.001$), and the mean latency to visit the correct cup (LM, $F_{1,26} = 6.24$; $p = 0.02$, Table 1, Fig 7). S+ individuals required more trials to reach the learning criterion (S+: $69.1 \pm 16.4$; S-: $41.4 \pm 20.6$), and took longer to visit the correct cup (S+: $3.4 \pm 1.0$ s; S-: $2.5 \pm 0.9$ s) compared with S- individuals.

## Discussion

This study aimed to improve our understanding of how variation in social motivation, used as a proxy for the broader sociability personality trait, relates to differences in sociocognitive performance. To this end, we took advantage of a unique model of Japanese quail selectively bred over several generations for divergent levels of social motivation: the S+ line, with high social motivation, and the S- line, with low social motivation. These quail were tested in a battery of sociocognitive tasks that varied in the type of social information provided and the social context: a gaze following task, a social buffering task, and a social discrimination learning task. Our results reveal that Japanese quail are capable of gaze following into the distance, and that this basic ability may constitute a widespread social capacity independent of individual

**Table 1. Means±standard deviations of the different variables recorded during the three social cognitive tasks (gaze following task, social buffering task, social discrimination learning task), presented by line (S+ and S-). A p-value in bold with \* indicates a significant difference (p≤0.05) and n.s. a non-significant difference.**

| Behavioural variable | S+ | S- | Model |
|---|---|---|---|
| **Gaze following task** | | | |
| Mean response rate during control trials (%) | No laser: 11.3±31.8<br>Laser: 13.1±33.9 | No laser: 22.5±41.9<br>Laser: 16.4±37.2 | Line: $\chi^2$=3.74; p=0.053<br>Condition: $\chi^2$=0.51; p=0.47<br>Interaction: n.s. |
| Mean response rate during test trials (%) | No laser: 21.4±41.1<br>Laser: 37.6±48.6 | No laser: 19.1±39.4<br>Laser: 38.1±48.7 | Line: $\chi^2$=0.00; p=1.00<br>Condition: $\chi^2$=23.09; **p =<0.001\***<br>Interaction: n.s. |
| **Social buffering task** | | | |
| Difference (s) | *Test 1* | | Line: F=5.50; **p=0.02\*** |
| | Alone: 70.8±51.9<br>Paired: 9.1±8.0 | Alone: 43.3±39.0<br>Paired: 15.5±32.0 | Condition: F=45.26; **p<0.001\*** |
| | *Test 2* | | Test: F=20.86; **p<0.001\*** |
| | Alone: 9.7±12.6<br>Paired: 0.9±6.1 | Alone: 8.6±7.9<br>Paired: 1.0±2.2 | Interactions: n.s. |
| | *Test 3* | | |
| | Alone: 101.6±30.9<br>Paired: 49.7±51.5 | Alone: 41.2±33.9<br>Paired: 27.3±41.3 | |
| | *Test 4* | | |
| | Alone: 46.3±46.8<br>Paired: 27.2±39.9 | Alone: 21.0±12.3<br>Paired: 14.1±36.2 | |
| Mean difference (s) | Alone (group 1): 57.1±21.5<br>Paired (group 3): 22.3±16.5 | Alone (group 2): 28.5±13.4<br>Paired (group 4): 14.5±14.1 | Group: F=11.64; **p<0.001\*** |
| **Social discrimination learning task** | | | |
| Total number of trials to reach the learning criterion | 69.1±16.4 | 41.4±20.6 | Line: F=15.71; **p<0.001\*** |
| Mean latency to visit the correct cup (s) | 3.4±1.0 | 2.5±0.9 | Line: F=6.24; **p=0.02\*** |

differences in social motivation. In contrast, line-dependent differences emerged in other tasks. S+ quail appear to be more sensitive and less able to cope with environmental changes, particularly under social isolation, in the social buffering task, whereas, unexpectedly, S- quail showed a greater tendency to rely on social information during the social discrimination learning task.

The results of the gaze following task suggest that Japanese quail are capable of following the gaze of a conspecific into the distance, particularly in an upward direction. These findings should nevertheless be interpreted with caution, given the inherent difficulty of inferring precise gaze direction from head orientation in this species. Despite this limitation, birds were significantly more likely to raise their heads towards a panel when a demonstrator had previously oriented towards it, compared to a condition in which no gaze cue was available. To our knowledge, this is the first experimental demonstration of gaze following in Japanese quail. Comparable response rates have been reported in other avian species, including corvids [55,56] such as ravens (*Corvus corax*, [57,58]), and non-corvid birds such as red junglefowl (*Gallus gallus*, [47]), Northern bald ibises (*Geronticus eremita*, [45]) and penguins (*Spheniscus demersus*, [46]), supporting the idea that gaze following is a widespread social capacity among birds. It has also been documented across a wide range of taxa, ranging from mammals to birds and reptiles [46,47].

Nevertheless, no differences were observed between S+ and S- individuals in the gaze following task, indicating that the two lines did not differ in their ability to co-orient with conspecifics' gaze, although this conclusion should be interpreted

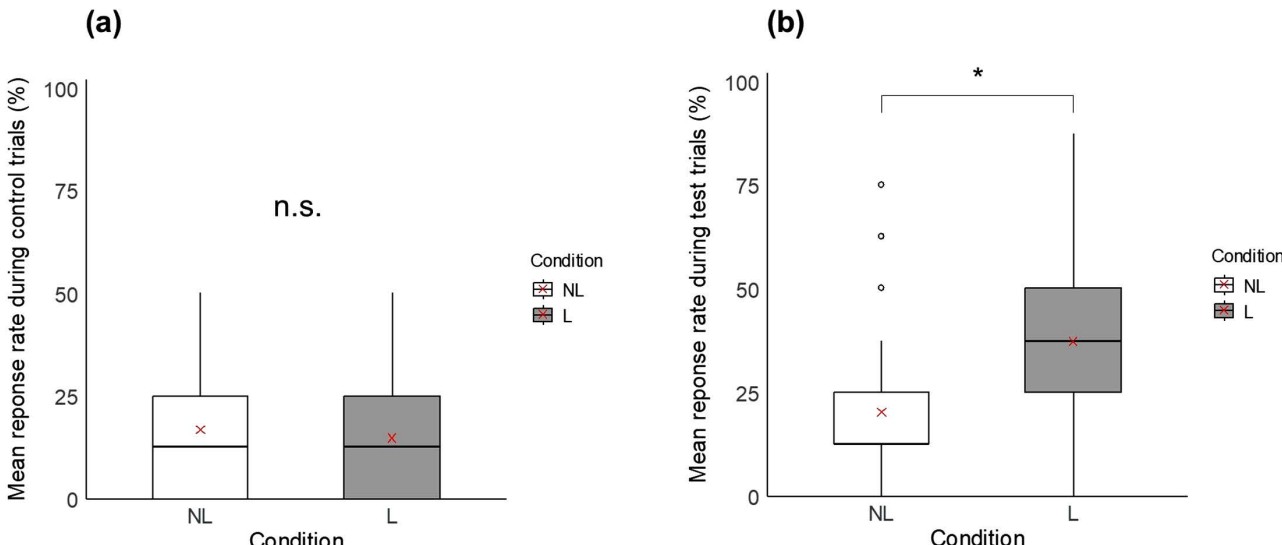

**Fig 5. Boxplots comparing in the gaze following task the mean response rate during (a) control trials, and (b) test trials, between the no laser condition (NL: white boxplots) and the laser condition (L: grey boxplots).** * indicates a significant difference (p ≤ 0.05) and n.s. a non-significant difference. Red crosses indicate mean values.

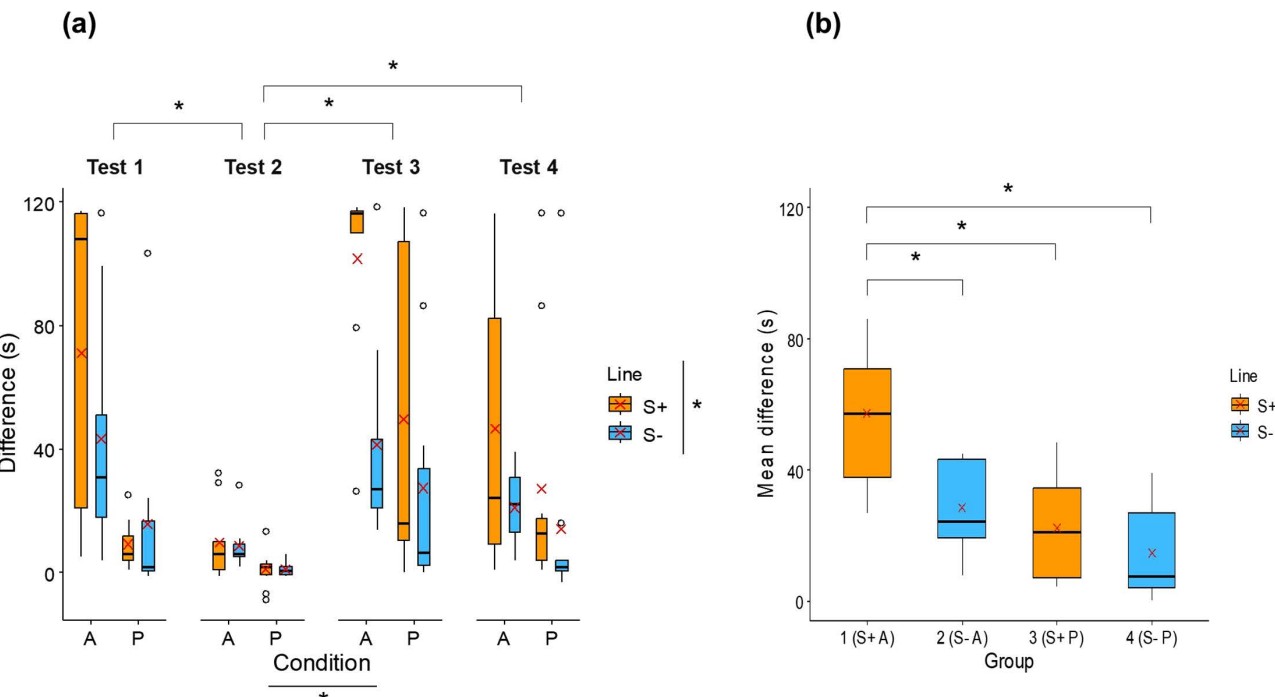

**Fig 6. Boxplots comparing in the social buffering task (a) the difference between the latency to reach the familiar feeder and the latency to reach the unfamiliar feeder (s), per line (S+: orange boxplots; S-: blue boxplots), per condition (A: alone; P: paired), and per test (1, 2, 3 and 4), and (b) the mean difference (s) across the four tests, between groups (1 = S+ alone, 2 = S- alone, 3 = S+ paired, 4 = S- paired).** * indicates a significant difference (p ≤ 0.05). Red crosses indicate mean values.

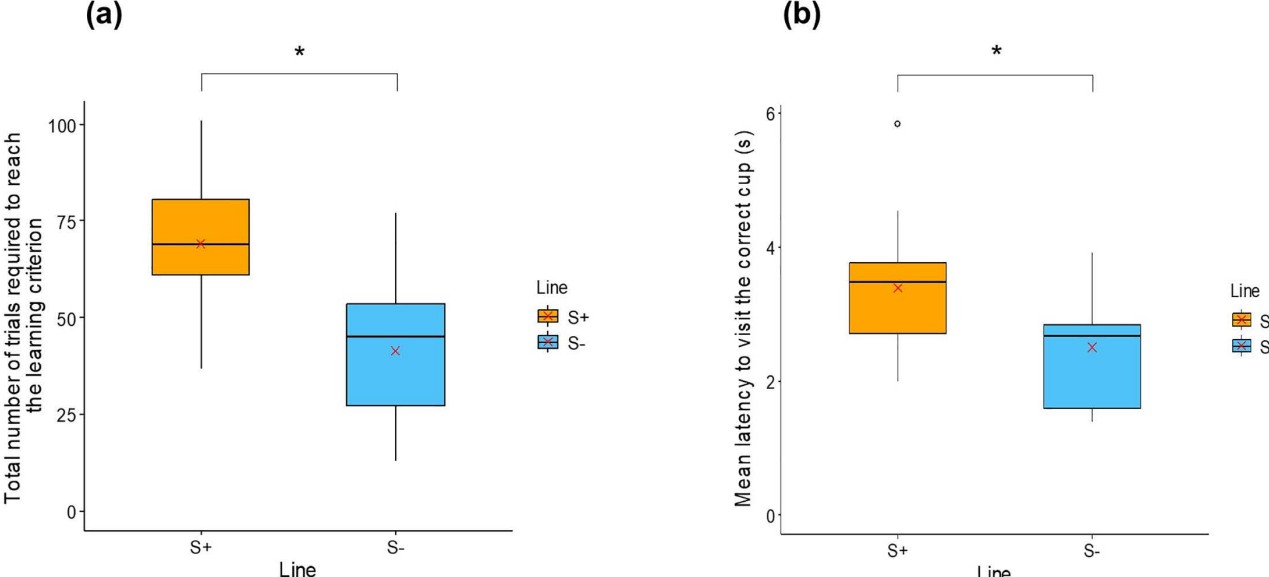

**Fig 7. Boxplots comparing in the social discrimination task (a) the total number of trials required to reach the learning criterion, and (b) the mean latency to visit the correct cup (s), between individuals with high social motivation (S+: orange boxplots) and those with low social motivation (S-: blue boxplots).** * indicates a significant difference (p ≤ 0.05). Red crosses indicate mean values.

cautiously as our method may not have been sensitive enough to detect subtle differences. This absence of divergence suggests that individual variation in social motivation does not modulate gaze following, at least in its most basic form. Indeed, the present task focused on gaze following into the distance, a low-level form of sociocognitive processing, which does not require prior expectations to find anything in the gaze direction, nor representations of the referentiality of the gaze, but is rather an adaptive reaction that drives attention to objects or events that could otherwise have been missed [47]. This interpretation is further supported by previous research showing that relationships between personality and cognition are more likely to emerge in tasks involving higher cognitive demand [3,26,28,59,60]. In contrast, simpler and more automatic responses are generally less sensitive to individual differences in personality traits. From a cognitive perspective, basic gaze following into the distance may therefore constitute a fundamental and broadly shared social capacity independent of variation in sociability level. Although many species exhibit basic gaze following, fewer demonstrate geometric gaze following, that is, an individual's ability to infer that another is looking at a target occluded by a barrier, causing the onlooker to reposition itself to see what the other is seeing [45–47]. This more sophisticated form of gaze following relies on more advanced cognitive processes, including visual perspective taking and spatial representation [45–47]. Investigating geometric gaze following may therefore provide a more demanding test of social cognitive abilities and could reveal an influence of social motivation, potentially uncovering differences between the S+ and S- lines that were not detected in the present task.

Regarding the social buffering task, our findings provide clear evidence for a social buffering effect: across both lines and tests, quail tested in pairs consistently approached the unfamiliar feeder more rapidly than quail tested alone. This indicates that the presence of a calm and experienced conspecific facilitated approach behaviour and alleviated the disruptive effect induced by environmental changes, supporting the notion that social companions can attenuate responses elicited by novel and potentially stressful situations [48,61,62]. A test effect was also observed, with Test 2 inducing noticeably less disturbance than the other three tests. This is likely because it represented the birds' second exposure to

a change in feeder colour, which may have facilitated habituation and consequently reduced their response. Furthermore, the white colour may have been less stressful than black [63].

In addition, the results revealed differences between S+ and S- individuals in their approach behaviour towards the unfamiliar feeder. Overall, S+ individuals took longer to approach the feeder than S- individuals, suggesting that they were generally more affected by changes in their environment. This result is in line with previous studies conducted on these divergent lines. In particular, a recent study has shown that individuals selected for high social motivation appear to be less flexible when faced with disruption. In a colour discrimination and reversal learning task, S+ quail persisted longer in their response to a previously rewarded coloured cup and showed impaired adjustment following contingency reversal, when reward was shifted to the alternative cup (Seressia et al., submitted). Similarly, in the present study, S+ individuals appeared to have greater difficulty coping with a disruption in their environment, particularly changes in feeder colour or surroundings. Thus, these results suggest that selection for high social motivation may be associated with heightened sensitivity and reduced adaptability to environmental changes.

A complementary analysis based on individual averages further revealed that the group composed of S+ quail tested alone differed from all other groups and appeared to be particularly affected by the environmental change. This pattern suggests that environmental novelty and disruption disproportionately affect highly sociable individuals when they are socially isolated. This interpretation is consistent with previous studies showing that S+ individuals exhibit greater behavioural reactivity under social isolation, displaying more stress-related and active social-seeking behaviours such as increased locomotor activity, vocalisations, and jumping [33,64]. These results further suggest that social buffering may be particularly important for highly socially motivated individuals, as their behavioural responses may become exaggerated in the absence of conspecifics, whereas in a social context their behaviour is more similar to that of less socially motivated individuals. However, this hypothesis requires further investigation, as it is primarily based on the outcomes of the global analysis and no significant interaction between line and testing condition was detected in the more comprehensive model. Taken together, these findings point to a complex interplay between sociability, social context (e.g., in social isolation or in group), and stress regulation.

Significant differences between the two lines also emerged in the social discrimination learning task. Contrary to our initial hypothesis, S- individuals showed superior performance, reaching the learning criterion more rapidly and approaching the correct cup faster than S+ individuals. The slightly shorter mean latency in S- birds is likely attributable to their higher success rate and, consequently, to their reduced number of incorrect choices requiring additional time to subsequently approach the second, correct cup. These results are consistent with previous studies, also reporting poorer performance in more sociable individuals in discrimination learning tasks. For instance, Lucon-Xiccato and Dadda [18] found that highly sociable guppies (*Poecilia reticulata*) showed poorer shoal size discrimination performance and were less efficient at choosing the larger shoal than less sociable individuals, despite the ecological relevance of this ability as a key antipredator strategy in social fish. Similarly, Nawroth et al. [19] reported that goats (*Capra hircus*) with lower sociability scores outperformed more sociable individuals in a visual discrimination task.

At first glance, these findings could be interpreted as reflecting superior learning abilities in less socially motivated individuals. However, closer examination of individual learning trajectories nuances this interpretation. A non-negligible proportion of S- quail reached the learning criterion extremely rapidly, including several individuals achieving maximal performance by the second training day. This observation suggests that S- individuals did not necessarily acquire the association more efficiently through gradual trial-and-error learning, but instead may have relied more spontaneously on social cues when making foraging decisions. Such performance was not observed in a previous study using the same discrimination learning procedure in Japanese quail, but with the cue being the color of the cup, suggesting that the effect depends on the social nature of the cue (Seressia et al., submitted). In contrast, S+ individuals may initially have relied more strongly on personal information at the beginning of the task, before having learned the discrimination rule. This interpretation aligns with the idea that individuals with lower social motivation may navigate their environment more

flexibly, adjusting their acquisition and use of available information according to context and associated benefits, whereas highly socially motivated individuals may rely more impulsively on self-generated information, even when social information would be advantageous (Seressia et al., submitted).

Alternative explanations for the poorer performance of S+ individuals should nevertheless be considered. One possibility is that S+ quail experienced higher stress levels during testing, leading to distraction and impaired performance, an interpretation that could be consistent with the results of the social buffering task reported above and with the hypothesis advanced by Nawroth et al. (2017) [19]. However, this explanation appears unlikely in the context of this task, as quail were extensively habituated to the experimental setup, showed comparable motivation during habituation, and were tested in the presence of a conspecific. Another possibility is that S- individuals were simply more socially motivated to approach the conspecific, rather than relying on it as a cue to guide their foraging choices, but this interpretation is difficult to reconcile with both the selection procedure and the robust body of evidence showing that S+ quail consistently display higher social motivation across a wide range of behavioural assays [29–33]. Moreover, based on behavioural observations during the task, individuals appeared to move directly towards the rewarded cup rather than seeking proximity to the conspecific, suggesting that performance differences are unlikely to reflect simple attraction to social partners. Finally, S- individuals may have perceived the conspecific more as a competitor in a foraging context, leading them to retrieve the feed on the conspecific's side more quickly. However, although feeding competition has not been specifically tested in these lines, previous studies involving them have not revealed any evidence of differences in competitive behaviour between S+ and S- individuals, and the experimental context of the present task did not involve direct competition, as the conspecific could not approach the feed item. Further empirical work will be needed to disentangle these possibilities.

Taken together, our findings show that the link between social motivation and social cognition is complex and strongly context-dependent. Highly socially motivated individuals were more sensitive to environmental changes, especially under social isolation, and less able to cope with them. In these situations, the presence of conspecifics acted as a strong social buffer and promoted more adaptive behaviour. However, in familiar and low-stress contexts, these individuals were less likely to rely spontaneously on social information. In contrast, less socially motivated individuals were more resilient to environmental change and more efficient at using social information in certain tasks, such as foraging challenges. While previous findings on the overall sociability of these quails suggest that their patterns of social relationships differ (for example, highly socially motivated individuals tend to be more behaviourally synchronized in groups and show no clear preference between familiar and unfamiliar conspecifics, [36,38]), further studies are needed to fully characterise their sociability beyond social motivation. In addition, the experimental selection procedure and/or the experimental conditions (such as group size) may have influenced other, unmeasured traits that could have contributed to the observed outcomes. These points should be kept in mind when interpreting the present results. Nevertheless, our findings highlight the importance of considering both individual differences and social context when studying links between personality and cognition. Cognitive performance cannot be understood independently of the social environment in which it occurs. Our results also open new avenues for research. In particular, the developmental origins of these patterns remain unknown. They may be present from hatching or emerge later, as individuals with different levels of social motivation could shape their social environments and, in turn, their cognitive demands. From an evolutionary perspective, the coexistence of different social strategies and context-dependent reliance on conspecifics may be adaptive and maintained by natural selection. Further research is needed to clarify these developmental and evolutionary mechanisms.

## Conclusions

In conclusion, inter-individual differences in social motivation, as a key component of sociability, shape social cognitive performance in a context-dependent manner. Selection on social motivation does not uniformly enhance or impair social cognition, but instead leads to differentiated strategies, notably in the acquisition and use of social information, that vary across tasks and situations. These findings highlight the complexity of the sociability-cognition relationship and emphasise

that cognitive performance cannot be fully understood without considering the interplay between personality traits, social context and environmental challenges. Beyond its theoretical relevance, this research may also be of applied interest, as it underscores the importance of social partners in a domesticated avian species.

## Supporting information

**S1 Data. Datasets used in the analyses.**
(XLSX)

## Acknowledgments

We are grateful to all members of the experimental unit PEAT of INRAE Nouzilly (France), especially the animal caretakers for practical assistance. We also thank Lucille Dumontier and Gaelle Lefort for statistical advice.

## Author contributions

**Conceptualization:** Jeanne SERESSIA, Léa Lansade, Christian Nawroth, Ludovic Calandreau, Vitor Hugo Bessa Ferreira.

**Data curation:** Jeanne SERESSIA, Alexia Morel.

**Formal analysis:** Jeanne SERESSIA, Alexia Morel.

**Funding acquisition:** Ludovic Calandreau, Vitor Hugo Bessa Ferreira.

**Investigation:** Jeanne SERESSIA, Alexia Morel.

**Methodology:** Jeanne SERESSIA, Alexia Morel, Fabien Cornilleau, Julie Lemarchand, Ludovic Calandreau, Vitor Hugo Bessa Ferreira.

**Project administration:** Ludovic Calandreau, Vitor Hugo Bessa Ferreira.

**Resources:** Fabien Cornilleau, Julie Lemarchand.

**Supervision:** Ludovic Calandreau, Vitor Hugo Bessa Ferreira.

**Validation:** Jeanne SERESSIA, Ludovic Calandreau, Vitor Hugo Bessa Ferreira.

**Visualization:** Jeanne SERESSIA.

**Writing – original draft:** Jeanne SERESSIA.

**Writing – review & editing:** Jeanne SERESSIA, Alexia Morel, Fabien Cornilleau, Julie Lemarchand, Léa Lansade, Christian Nawroth, Ludovic Calandreau, Vitor Hugo Bessa Ferreira.

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
