## [Decision Letter · Decision Letter 0]

26 Mar 2026

PONE-D-26-08810Japanese quail (Coturnix japonica) selected for high social motivation rely on conspecifics for buffering but not learning in sociocognitive tasksPLOS One

Dear Dr. SERESSIA,

Thank you for submitting your manuscript to PLOS ONE. After careful consideration, we feel that it has merit but does not fully meet PLOS ONE’s publication criteria as it currently stands. Therefore, we invite you to submit a revised version of the manuscript that addresses the points raised during the review process.

We look forward to receiving your revised manuscript.

Kind regards,

Arda Yildirim, Ph.D.

Academic Editor

PLOS One

**Journal Requirements:**

1. When submitting your revision, we need you to address these additional requirements. Please ensure that your manuscript meets PLOS ONE's style requirements, including those for file naming. The PLOS ONE style templates can be found at https://journals.plos.org/plosone/s/file?id=wjVg/PLOSOne_formatting_sample_main_body.pdf and https://journals.plos.org/plosone/s/file?id=ba62/PLOSOne_formatting_sample_title_authors_affiliations.pdf 2. To comply with PLOS One submissions requirements, in your Methods section, please provide additional information regarding the experiments involving animals and ensure you have included details on (a) methods of sacrifice, (b) methods of anesthesia and/or analgesia, and (c) efforts to alleviate suffering. 3. We note that Figure 2 in your submission contain copyrighted images. All PLOS content is published under the Creative Commons Attribution License (CC BY 4.0), which means that the manuscript, images, and Supporting Information files will be freely available online, and any third party is permitted to access, download, copy, distribute, and use these materials in any way, even commercially, with proper attribution. For more information, see our copyright guidelines: http://journals.plos.org/plosone/s/licenses-and-copyright. We require you to either present written permission from the copyright holder to publish these figures specifically under the CC BY 4.0 license, or remove the figures from your submission: a. You may seek permission from the original copyright holder of Figure 2 to publish the content specifically under the CC BY 4.0 license.  We recommend that you contact the original copyright holder with the Content Permission Form (http://journals.plos.org/plosone/s/file?id=7c09/content-permission-form.pdf) and the following text:“I request permission for the open-access journal PLOS ONE to publish XXX under the Creative Commons Attribution License (CCAL) CC BY 4.0 (http://creativecommons.org/licenses/by/4.0/). Please be aware that this license allows unrestricted use and distribution, even commercially, by third parties. Please reply and provide explicit written permission to publish XXX under a CC BY license and complete the attached form.” Please upload the completed Content Permission Form or other proof of granted permissions as an "Other" file with your submission.  In the figure caption of the copyrighted figure, please include the following text: “Reprinted from [ref] under a CC BY license, with permission from [name of publisher], original copyright [original copyright year].” b. If you are unable to obtain permission from the original copyright holder to publish these figures under the CC BY 4.0 license or if the copyright holder’s requirements are incompatible with the CC BY 4.0 license, please either i) remove the figure or ii) supply a replacement figure that complies with the CC BY 4.0 license. Please check copyright information on all replacement figures and update the figure caption with source information. If applicable, please specify in the figure caption text when a figure is similar but not identical to the original image and is therefore for illustrative purposes only. 4. Please include captions for your Supporting Information files at the end of your manuscript, and update any in-text citations to match accordingly. Please see our Supporting Information guidelines for more information: http://journals.plos.org/plosone/s/supporting-information. 5. If the reviewer comments include a recommendation to cite specific previously published works, please review and evaluate these publications to determine whether they are relevant and should be cited. There is no requirement to cite these works unless the editor has indicated otherwise.

**Additional Editor Comments:**

Thank you for submitting your manuscript. The reviewers highlighted several important issues that need to be addressed, particularly regarding methodological clarity, data interpretation, and the strength of the conclusions. The discussion should be improved by better integrating relevant literature and avoiding overinterpretation of the results. Therefore, substantial revision is required, and the manuscript will need to undergo re-assessment following revision. Regards, Arda Yıldırım

Reviewers' comments:

Reviewer's Responses to Questions

**Comments to the Author**

1. Is the manuscript technically sound, and do the data support the conclusions?

Reviewer #1: Yes

Reviewer #2: Partly

Reviewer #3: Yes

2. Has the statistical analysis been performed appropriately and rigorously? 

Reviewer #1: Yes

Reviewer #2: Yes

Reviewer #3: Yes

3. Have the authors made all data underlying the findings in their manuscript fully available?

Reviewer #1: Yes

Reviewer #2: Yes

Reviewer #3: Yes

4. Is the manuscript presented in an intelligible fashion and written in standard English?

Reviewer #1: Yes

Reviewer #2: Yes

Reviewer #3: Yes

5. Review Comments to the Author

**Reviewer #1:** The introduction is well written and nicely focuses on the main goal of the study. I enjoy reading it.

Methods clearly state all the procedures and details of the test nicely accompanied by figures further helping to understand the timeline and the procedures.

Discussion of the social discrimination task. One more alternative explanation might be considered and that is that for S- individuals the conspecific represents or is perceived more like competitor and thus they need to quickly extract the food before the conspecific can potentially reach it. I am just curious if the authors might think this is possible or not. From their experience with behavior of the two lines.

Overall, the discussion is again well written and includes alternative explanations like the stress sensitivity, which I appreciate.

On several places in the manuscript the authors state that the S+ individuals are more sensitive to environmental change and less flexible in their response. I would agree with the first part of the statement. As we see, there is a clear difference in behavior when the feeder changes. But the fact that we see the different behavior contradicts that they are less flexible. At least in my understanding of the term flexible is that individuals are changing their behavior based on the conditions and the “nonflexibility” means they behave all the same under different conditions. So, I would say they are more sensitive and less able to cope with the changes of conditions (not less flexible].

**Reviewer #2:** This study investigates differences in performance on social cognitive tasks of quail strains bred based on different levels of social motivation. The study is novel. The manuscript is well written. My comments aim to help identify areas where additional clarification would provide additional clarity and transparency. A few topics that warrant further explanation or discussion are also highlighted. These include a priori presentation of hypotheses and predictions, further discussion of study limitations and exploration of alternative explanations. Below are some specific comments for the authors’ consideration.

Abstract: Clearly written

Introduction: Clearly written and relevant. Explicitly stating the hypotheses and predicted outcomes of each test in the introduction would help further clarify the rest of the manuscript.

Materials & Methods: Mostly clear. I have a few comments for the authors to consider:

Throughout: Gaze task should be referred to as modified gaze task since gaze is not actually measured, and it is unclear where the focal bird is actually looking.

Throughout: Who scored the behaviors? Was intra and/or inter rater reliability calculated?

LN 91: Important to discuss that social motivation may drive sociability but the to are not always correlated.

LN 120: The heading title seems to be a partial sentence

LN 124: The treadmill tests, described in lines 127-137, was used to determine which birds were mated. Was mating random within groups starting with the 46th generation?

LN 176-179: This implies that one pen (one strain) was tested on any given morning and the other strain that afternoon. Is that correct?

LN 214: Did all of the tested birds meet these criteria by the end of the 12 habituation days?

LN 250-251: Based on the description it appears that all movements of the head or eye toward any part of the front partition or toward any part of the back wall were included. Given the position of the eyes this seems to cover much of the possible head movement. This uncertainty surrounding where the bird was actually looking is illustrated in the photo presented as the middle left panel of Figure 2, which shows the head oriented toward the side wall (though the eye may be looking back). Were there any criteria or the angle of the head or eye in respect to the location of the panel? Is it the case that pretty much any movement upwards was counted as long as it was sufficiently pronounced?

LN 250: Was reliability of observations quantified (inter/intra rater reliability)?

LN 356: Was the amount of mealworms presented standardized across training and testing sessions?

LN 374-276: I am not convinced that this would prevent the quail from using its sense of smell to find the mealworms. I would imagine that some amount of mealworms in the open cup would give off a stronger odor than a single one placed under a false bottom. I recommend revising the text to indicate that this was an effort to decrease likelihood that smell guided the choice, and discussing this limitation.

Statistical analyses: Clearly explained. Sufficient detail provided. However, having predictions explicitly stated earlier in the manuscript would help clarify why these models were chosen (e.g. why the control and test treatments were not compared in the modified gaze test).

LN 418: Were any corrections made for multiple testing on same date?

Results:

LN 438: The results refer to an expected outcome; it would be helpful to have predicted outcomes stated in the introduction.

LN 459: Change “et” to “and”

Discussion:

LN 507 & LN 513-522: Please see my comments about gaze not being measured. Based on the provided information, it is difficult to conclude whether or not the birds were following the demonstrator’s gaze. A discussion of assumptions about what the birds are looking at is warranted.

LN 523: On the same topic, is it possible that the data collection method simply was not sensitive enough to identify any differences given that it is uncertain where the birds were looking?

LN 512: was there any evidence that these birds simply learned more quickly overall?

LN 543-545: Could it have been that the ST birds didn’t just serve as social buffers, but were motivating the other birds to approach due to resource competition since the ST birds were allowed to feed first and birds were tested after some level of feed deprivation?

LN 612: It would have been interesting for the authors to explore alternative explanations. While the two strains were bred based on a test of social motivation, is it not possible that other changes occurred simultaneously which may have driven the observed results?

Figures & Table: Provide helpful visual representation of the methods and results.

**Reviewer #3:** This article explores the relationship between social behaviour – and more specifically, social motivation – and social cognitive abilities in the Japanese quail. These abilities are assessed using three tests: the gaze-following task, the social buffering task, and a social discriminationlearning task.

The question addressed in this study is highly interesting as it aims to understand the link between sociality and social cognition, in relation to the social intelligence hypothesis. This species has been little studied in relation to this issue and, as such, this study provides a valuable comparative insight into the evolutionary processes linking social complexity and social cognition.

Introduction/

The introduction is very clear. However, the authors could provide more detail on the various components or types of social abilities that can be analyzed in species, and their different levels of complexity. This would help to link these various abilities to the tests performed in the study. Indeed, these tests are described as complementary, but this is not explained. Similarly, the authors could link these social cognitive abilities to potential different neuro-biological processes.

Furthermore, in addition to the existence of strains selected for social motivation, the authors could also justify the choice of the bird model used in light of its lifestyle.

Methods/

Line 185. ST birds were used to produce the demonstrators and are reared with S+ or S- individuals. However, during the tests, the ST demonstrators, for example of a focal S- individual, comes from the pen of S+ birds. Even if I understand the need to control for social bonds between individuals, does the fact that the control birds are reared with a particular strain not lead to the development of a behavioural phenotype linked to that strain, and thus, could influence birds’ interaction and results.

Moreover, the learning tasks involved a significant duration of experimentation, which must have encompassed the birds’ sexual development and thus the onset of egg-laying in the females. Was the reproductive status of the females identical between the S+ and S- groups at the time of each test? The physiological status of individuals can alter their social behaviour, potentially leading to more aggressive behaviour. Was this type of interaction observed and recorded during the tests? How might sexual development have influenced your results?

Results/

Line 438. Even though there is no line effect on the gaze following task, it would be interesting to show the data for each line separately in Figure 5, particularly as the probability is not so far from the significance level.

Line 453. The birds’ response in Test 2 of the social buffering task appears very different from the other tests; Indeed, there is little variation in behaviour. This corresponds to the colour white. Can the authors explain this difference for this test? Is white less anxiety-inducing for quail?

With regard to the results on the social learning task, might the observed effects have been influenced by the size of the groups during the experiment? Indeed, the birds were reared in large groups, which could have affected the nature of their social relationships and thus potentially their social cognition. Indeed, previous studies on quail strains show an increase in agonistic behaviour as group size increases, as well as weaker social bonds. The authors could discuss their results in light of a potential group size effect.

6. PLOS authors have the option to publish the peer review history of their article (what does this mean?). If published, this will include your full peer review and any attached files.

Reviewer #1: No

Reviewer #2: No

Reviewer #3: No

---

## [Author Response · Author response to Decision Letter 1]

7 Apr 2026

April 3, 2026

Subject: Submission of revised paper PONE-D-26-08810

Dear Arda Yildirim,

Editor, PLOS One

Thank you for your email of March 26, 2026, conveying your comments and those of the three reviewers regarding our manuscript entitled “Japanese quail (Coturnix japonica) selected for high social motivation rely on conspecifics for buffering but not learning in sociocognitive tasks”. We have carefully considered all comments and have revised the manuscript accordingly. We believe that these revisions have strengthened the clarity and rigor of our work. Please find below a detailed, point-by-point response to each comment. For clarity, the Editor’s and Reviewers’ comments are reproduced in full (highlighted in grey), followed by our responses. All changes made to the manuscript are highlighted in green in the revised version. We hope that the revised manuscript now meets the standards required for publication in PLOS One.

Yours sincerely,

Author and co-authors

Journal Requirements :

Done as requested.

2. To comply with PLOS One submissions requirements, in your Methods section, please provide additional information regarding the experiments involving animals and ensure you have included details on (a) methods of sacrifice, (b) methods of anesthesia and/or analgesia, and (c) efforts to alleviate suffering.

The Materials and methods section, particularly the Ethics statement, has been revised and completed as requested (L141-154).

3. We note that Figure 2 in your submission contain copyrighted images. All PLOS content is published under the Creative Commons Attribution License (CC BY 4.0), which means that the manuscript, images, and Supporting Information files will be freely available online, and any third party is permitted to access, download, copy, distribute, and use these materials in any way, even commercially, with proper attribution.

The images were taken by the authors and originate directly from the experimental tests.

Author (Jeanne Seressia) and co-authors grant permission to the open-access journal PLOS ONE to publish Figure 2 of the manuscript entitled “Japanese quail (Coturnix japonica) selected for high social motivation rely on conspecifics for buffering but not learning in sociocognitive tasks” under the Creative Commons Attribution License (CCAL) CC BY 4.0.

4. Please include captions for your Supporting Information files at the end of your manuscript, and update any in-text citations to match accordingly.

Done as requested : S1 Data (L899).

Done as requested.

Additional Editor Comments :

Thank you for submitting your manuscript. The reviewers highlighted several important issues that need to be addressed, particularly regarding methodological clarity, data interpretation, and the strength of the conclusions. The discussion should be improved by better integrating relevant literature and avoiding overinterpretation of the results. Therefore, substantial revision is required, and the manuscript will need to undergo re-assessment following revision. Regards, Arda Yıldırım

Response to Reviewer 1 :

We highly appreciate the reviewer’s comments. The followings are our point-by-point responses :

Reviewer #1: The introduction is well written and nicely focuses on the main goal of the study. I enjoy reading it.

Methods clearly state all the procedures and details of the test nicely accompanied by figures further helping to understand the timeline and the procedures.

Discussion of the social discrimination task. One more alternative explanation might be considered and that is that for S- individuals the conspecific represents or is perceived more like competitor and thus they need to quickly extract the food before the conspecific can potentially reach it. I am just curious if the authors might think this is possible or not. From their experience with behavior of the two lines.

We thank the reviewer for this very interesting suggestion, and we indeed cannot rule out this possibility. Based on our experience with these lines, we have no evidence to suggest that individuals from the S- line might perceive their conspecifics more as competitors compared with those from the S+ line, and testing this hypothesis more specifically could be interesting in future studies. Moreover, the experimental context of the social discrimination learning task did not involve direct competition for access to the feed resource, as the conspecifics could not reach the feed item, including during the habituation phase during which conspecifics were always kept in their compartment. We nevertheless included this alternative explanation to the manuscript :

(L672-678) Finally, S- individuals may have perceived the conspecific more as a competitor in a foraging context, leading them to retrieve the feed on the conspecific’s side more quickly. However, although feeding competition has not been specifically tested in these lines, previous studies involving them have not revealed any evidence of differences in competitive behaviour between S+ and S- individuals, and the experimental context of the present task did not involve direct competition, as the conspecific could not approach the feed item. Further empirical work will be needed to disentangle these possibilities.

Overall, the discussion is again well written and includes alternative explanations like the stress sensitivity, which I appreciate.

On several places in the manuscript the authors state that the S+ individuals are more sensitive to environmental change and less flexible in their response. I would agree with the first part of the statement. As we see, there is a clear difference in behavior when the feeder changes. But the fact that we see the different behavior contradicts that they are less flexible. At least in my understanding of the term flexible is that individuals are changing their behavior based on the conditions and the “nonflexibility” means they behave all the same under different conditions. So, I would say they are more sensitive and less able to cope with the changes of conditions (not less flexible].

The reviewer is absolutely right. We have revised the manuscript accordingly :

(L557-560) S+ quail appear to be more sensitive and less able to cope with environmental changes, particularly under social isolation, in the social buffering task, whereas, unexpectedly, S- quail showed a greater tendency to rely on social information during the social discrimination learning task.

(L680-681) Highly socially motivated individuals were more sensitive to environmental changes, especially under social isolation, and less able to cope with them.

Response to Reviewer 2 :

We highly appreciate the reviewer’s comments. The followings are our point-by-point responses :

Reviewer #2: This study investigates differences in performance on social cognitive tasks of quail strains bred based on different levels of social motivation. The study is novel. The manuscript is well written. My comments aim to help identify areas where additional clarification would provide additional clarity and transparency. A few topics that warrant further explanation or discussion are also highlighted. These include a priori presentation of hypotheses and predictions, further discussion of study limitations and exploration of alternative explanations. Below are some specific comments for the authors’ consideration.

Abstract: Clearly written

Introduction: Clearly written and relevant. Explicitly stating the hypotheses and predicted outcomes of each test in the introduction would help further clarify the rest of the manuscript.

As requested, we have specified the hypotheses and expected outcomes for each test in the introduction of the revised manuscript :

(L131-133) We hypothesized that S+ individuals would be more likely to follow the gaze of conspecifics, show greater modulation of stress responses by social context, and rely more readily on social information, resulting in faster association in the social discrimination learning task.

Materials & Methods: Mostly clear. I have a few comments for the authors to consider:

Throughout: Gaze task should be referred to as modified gaze task since gaze is not actually measured, and it is unclear where the focal bird is actually looking.

We thank the reviewer for this important comment. As correctly pointed out, determining precisely where a bird is looking based solely on head orientation may be challenging, particularly given the lateral position of the eyes in quail and the absence of a visible scleral contrast that could help refine gaze direction estimates. In our study, head movements were scored as gaze following responses only when they were salient, and focusing on instances where the focal bird tilted its head towards the panel, or towards the wall behind, and/or markedly elevated its head. Nevertheless, we fully agree that some uncertainty inevitably remains regarding the exact direction of gaze. Importantly, this operational definition follows commonly used criteria in gaze following studies in birds, where head orientation is typically used as a proxy for gaze direction (Loretto et al., 2010; Nawroth et al., 2017; Zeiträg et al., 2023, cited in the manuscript L228). Moreover, it is inherently difficult to determine how the focal bird itself perceived the demonstrator’s gaze, which may not correspond precisely to a single spatial point but rather to a broader area of the testing arena. Defining a strict criterion based on a specific head-angle threshold would also have been very challenging and we therefore focused on clear head-orienting movements to identify gaze following responses. Our results showed that quail produced markedly more clear head-orienting responses when the conspecific had previously raised its head than when it had not, supporting the interpretation that these responses could reflect gaze following behaviour. We suggest retaining the term “gaze following task”, as the protocol follows a widely used procedure for assessing gaze following in birds. However, we have added in the Discussion section a note advising caution in interpreting the results, given the inherent difficulty of assessing precise gaze direction in this species :

(L561-566) The results of the gaze following task suggest that Japanese quail are capable of following the gaze of a conspecific into the distance, particularly in an upward direction. These findings should nevertheless be interpreted with caution, given the inherent difficulty of inferring precise gaze direction from head orientation in this species. Despite this limitation, birds were significantly more likely to raise their heads towards a panel when a demonstrator had previously oriented towards it, compared to a condition in which no gaze cue was available.

Throughout: Who scored the behaviors? Was intra and/or inter rater reliability calculated?

Data were collected by two researchers (JS and AM). For each test, behaviours were scored by a single observer to ensure within-test consistency. Prior to data collection, both observers underwent a training period, during which behaviours were discussed and operational definitions were refined until agreement was reached. Data collection was then conducted independently. Moreover, whenever possible, behaviours were first recorded live and then systematically verified using video recordings to confirm scoring accuracy. The three tasks relied on salient variables (clear head movements in the gaze following task (L296-299), latency to reach the feeder and place the head above it in the social buffering task (L373-377), and the binary success outcome together with latency to visit the correct cup and consume the mealworm in the social discrimination task (L420-423, L427-429), and the results showed strong effects. We have clarified this information in the revised manuscript :

(L222-225) Data were collected by two researchers (JS and AM). For each test, behaviours were scored by a single observer to ensure within-test consistency. Prior to data collection, both observers underwent a training period, during which behaviours were discussed and operational definitions were refined until agreement was reached. Data collection was then conducted independently.

LN 91: Important to discuss that social motivation may drive sociability but the to are not always correlated.

The reviewer is absolutely right. Social motivation can be an important component of sociability, and social reinstatement is very often used as a proxy to measure this broader trait, however it does not necessarily demonstrate that individuals are more sociable in a general sense, even though individuals from our lines appear to differ more broadly in their social behaviours beyond social motivation alone (L90-103). Taking this into account, we discussed our results in the manuscript primarily in relation to social motivation (e.g., L46-50, L576-577, L614-616, L679-680, L704-708). Nevertheless, to emphasise this point further, we have now explicitly acknowledged it again as a limitation of our study at the end of the discussion section :

(L686-694) While previous findings on the overall sociability of these quails suggest that their patterns of social relationships differ (for example, highly socially motivated individuals tend to be more behaviourally synchronized in groups and show no clear preference between familiar and unfamiliar conspecifics, [36,38]), further studies are needed to fully characterise their sociability beyond social motivation. In addition, the experimental selection procedure and/or the experimental conditions (such as group size) may have influenced other, unmeasured traits that could have contributed to the observed outcomes. These points should be kept in mind when interpreting the present results. Nevertheless, our findings highlight the importance of considering both individual differences and social context when studying links between personality and cognition.

LN 120: The heading title seems to be a partial sentence

The heading title has been rephrased (L156).

LN 124: The treadmill tests, described in lines 127-137, was used to determine which birds were mated. Was mating random within groups starting with the 46th generation?

Yes, that is correct. In the first 45 generations, the treadmill test was conducted to identify individuals at the extremes of social motivation. Those with the highest social motivation were used as breeders for the S+ line in the following generation, while those with the lowest social motivation were used as breeders for the S- line. From the 46th generation onwards, the selection test was no longer performed, and breeders were chosen randomly within each line to produce the next generation. We have revised this section in the manuscript to make the procedure clearer :

(L159-166) Selection was originally performed using a treadmill test until the 45th generation. This test was conducted to identify individuals at the extremes of social motivation: those with the highest social motivation were used as breeders for the S+ line in the following generation, while those with the lowest social motivation were used as breeders for the S- line. Then, from the 46th generation onwards, the selection test was no longer performed and breeders were chosen randomly within each line to produce the next generation. However, the treadmill test has been carried out approximately every three years on a random sample of birds from both lines to ensure that the divergence was maintained.

LN 176-179: This implies that one pen (one strain) was tested on any given morning and the other strain that afternoon. Is that correct?

Actually, that was not the case, and we thank the reviewer for the opportunity to clarify this point. Half of the S+ and S- individuals were tested in the morning and the other half in the afternoon

---

## [Decision Letter · Decision Letter 1]

30 Apr 2026

Japanese quail (Coturnix japonica) selected for high social motivation rely on conspecifics for buffering but not learning in sociocognitive tasks

PONE-D-26-08810R1

Dear Dr. SERESSIA,

We’re pleased to inform you that your manuscript has been judged scientifically suitable for publication and will be formally accepted for publication once it meets all outstanding technical requirements.

Kind regards,

Arda Yildirim, Ph.D.

Academic Editor

PLOS One

Additional Editor Comments (optional):

Dear Authors,

Thank you for your revised submission. The revisions have satisfactorily addressed the majority of the reviewers' comments and concerns. I am pleased to inform you that your manuscript is now recommended for acceptance and publication. Best regards, Arda Yıldırım

Reviewers' comments:

Reviewer's Responses to Questions

**Comments to the Author**

1. If the authors have adequately addressed your comments raised in a previous round of review and you feel that this manuscript is now acceptable for publication, you may indicate that here to bypass the “Comments to the Author” section, enter your conflict of interest statement in the “Confidential to Editor” section, and submit your "Accept" recommendation.

Reviewer #3: All comments have been addressed

2. Is the manuscript technically sound, and do the data support the conclusions?

Reviewer #3: Yes

3. Has the statistical analysis been performed appropriately and rigorously? 

Reviewer #3: Yes

4. Have the authors made all data underlying the findings in their manuscript fully available?

Reviewer #3: Yes

5. Is the manuscript presented in an intelligible fashion and written in standard English?

Reviewer #3: Yes

6. Review Comments to the Author

Reviewer #3: All comments have been addressed

All comments have been addressed

All comments have been addressed

All comments have been addressed

7. PLOS authors have the option to publish the peer review history of their article (what does this mean?). If published, this will include your full peer review and any attached files.

Reviewer #3: No

---

## [Editor Report · Acceptance letter]

PONE-D-26-08810R1

PLOS One

Dear Dr. SERESSIA,

I'm pleased to inform you that your manuscript has been deemed suitable for publication in PLOS One. Congratulations! Your manuscript is now being handed over to our production team.

Kind regards,

on behalf of

Prof. Dr. Arda Yildirim

Academic Editor

PLOS One